# Qualitative and Quantitative Real-Time PCR Methods for Assessing False-Positive Rates in Genetically Modified Organisms Based on the Microbial-Infection-Linked *HPT* Gene

**DOI:** 10.3390/ijms231710000

**Published:** 2022-09-02

**Authors:** Yunjing Li, Fang Xiao, Chao Zhai, Xiaofei Li, Yuhua Wu, Hongfei Gao, Jun Li, Shanshan Zhai, Biao Liu, Gang Wu

**Affiliations:** 1Key Laboratory of Agricultural Genetically Modified Organisms Traceability, Ministry of Agriculture and Rural Affairs, Oil Crops Research Institute of Chinese Academy of Agricultural Science/Supervision and Test Center (Wuhan) for Plant Ecological Environment Safety, Ministry of Agriculture and Rural Affairs, Wuhan 430062, China; 2School of Life Sciences, Hubei University, Wuhan 430062, China; 3Nanjing Institute of Environmental Sciences, Ministry of Ecology and Environment of the People’s Republic of China, Nanjing 210042, China

**Keywords:** *HPT* gene, GMO, microbial infection, qualitative PCR, qPCR, detection methods

## Abstract

The hygromycin phosphotransferase (*HPT*) gene as a selective marker is normally used in screening tests as a first step in detecting and quantifying genetically modified organisms (GMOs) in seeds, food, and feed materials. Nevertheless, if researchers only focus on the *HPT* gene, it is difficult to distinguish genetically modified (GM) crops from microbial infection, leading to miscalculation of the rate of GM materials in a given sample set. Here, we cloned the 7259 bp sequence carrying the *HPT* gene from soybean sprouts using the genome walking strategy. BLAST analysis revealed that this sequence was derived from plasmids naturally occurring in microorganisms, such as *Escherichia coli*, *Klebsiella pneumoniae* or *Salmonella* sp. Using the reconstructed plasmid pFP-hpt, qualitative PCR and quantitative real-time PCR (qPCR) methods were established, and 261 bp and 156 bp products were produced. The specificity of these assays was assessed against related pFP-hpt plasmids, plant species with important agronomic traits, and GM crops containing the *HPT* gene. No unexpected results were observed between samples using these qualitative PCR and qPCR methods. The sensitivity of this qualitative PCR assay was determined at 20 copies, while the limit of detection (LOD) and limit of quantification (LOQ) of qPCR were both 5 copies per reaction. Our in-house validation indicated that the amplification efficiency, linearity, and repeatability of this qPCR assay were in line with performance requirements. Furthermore, a qualitative and quantitative duplex PCR showed high reliability for the simultaneous detection of the *HPT* gene in a plant sample and environmental micro-organisms harboring the *HPT* gene in one PCR reaction. These qualitative PCR and qPCR assays were able to differentiate between plants infected with *E. coli* harboring the *HPT* gene from GM plants, indicating that these two methods are broadly applicable for routine GMO testing.

## 1. Introduction

Due to the annual increase in the variety and number of genetically modified (GM) plants, monitoring and management systems for seeds, food, and feed products for the presence of GMOs are enforced in many countries worldwide, which require reliable and cost-effective detection methods. DNA analysis based on polymerase chain reaction (PCR) is routinely performed to assess the presence or absence of GMOs. Several common genetic elements present in many GMOs, such as the *CaMV35S* promoter derived from the cauliflower mosaic virus and the *NOS* terminator derived from *Agrobacterium tumefaciens*, as well as selectable markers including the *pat*/*bar* gene, the *HPT* gene, and the neomycin phosphotransferase type II (*NPTII*) gene from *Bacillus thuringiensis*, are usually selected as detection targets in GMO screening [1,2,3]. These elements are commonly derived from naturally occurring viruses, bacteria, fungi, and plants. A positive detection result with an element screening assay is sometimes but not always a confirmation of the presence of transgenic DNA in a sample. In China, the state has adopted a zero-tolerance policy for the management of GMOs that have not yet been approved for commercial use. Once the false-positive results of element-screening tests are adopted, the producers or operators will be penalized according to the Agricultural GMO Safety Management Regulations. The laboratory that performed the testing will also be subject to penalties in cases of fraud or negligence in judging the results. Therefore, a specific test method is required to avoid false-positive results due to the presence of naturally occurring organisms that happen to be the donor organism for the genetic element being assessed.

PCR-based methods, such as conventional PCR, qPCR, and multiplex quantitative PCR, have been demonstrated to be able to detect cauliflower mosaic virus (CaMV) and figwort mosaic virus (FMV) DNA to differentiate between GMOs and the natural occurrence of transgenic elements of the *CaMV35S* and *FMV35S* promoters, respectively [4,5,6,7,8,9]. The selection of selectable marker genes is one of the most important factors for the success of plant genetic transformation and positive offspring screening [10]. As a commonly used selection marker, the *HPT* gene is widely used in the transformation of monocot plants, including rice [11], maize [12], wheat [13,14], and barley [15]. Several GM plants harboring the *HPT* gene are in the stage of productivity test or have been approved as imported processed raw materials in China, such as Kefeng 6, Kemingdao, or COT102. However, many more plants with the *HPT* gene are currently being experimentally investigated, which are considered unauthorized GMOs. The *HPT* gene is derived from *E. coli*, which is universally present in nature and can readily infect genetically modified seeds, food, and feed, especially for the rice samples. It is therefore probable that food, feed or seed materials are contaminated by the *HPT* gene from *E. coli*, which can cause false-positive results in PCR tests targeting the *HPT* element as a GM marker. The *HPT* gene was expressed by the *CaMV35S* promoter and *NOS* terminator in many transgenic plants. In daily multiple-element-screening tests, transgenic testing laboratories encountered situations where only the *HPT* gene was detected, and due to the lack of an *HPT* gene false-positive detecting method, it was difficult for them to make a correct judgment.

Since qualitative PCR and qPCR methods that can distinguish GMOs from the natural occurrence of transgenic elements for the *HPT* gene have not been published, there is an urgent requirement by GMO testing laboratories for establishing *HPT* gene false-positive test systems. In the present study, a 7259 bp sequence including the *HPT* gene, which was highly homologous to the plasmid sequences of *Escherichia coli*, *Klebsiella pneumoniae*, and *Salmonella* sp., was cloned from soybean sprouts. Qualitative PCR and qPCR methods are presented that allow the detection of plasmid-DNA-specific regions between the *HPT* gene and its 5′-flanking sequence. The qualitative PCR assay targets a 261 bp sequence of the plasmid-DNA-specific region, and qPCR assay detecting fragment is shortened to 156 bp within the 261 bp region. This newly developed assay can be used to verify that a positive *HPT* screening result is caused by the presence of *E. coli* carrying *HPT* and not due to a GM plant event in a given sample with good sensitivity and specificity.

## 2. Results

### 2.1. Routine Detection of GMO in Soybean Sprouts

Soybean sprouts purchased from a local market were first investigated by screening PCR in routine analysis. We chose them because they had tested positive for the *HPT* gene found in GM crops [16]. However, no other GMO elements could be detected using soybean event-specific PCR methods, as well as other screening elements (*CaMV35S* promoter, *NOS* terminator, *Cp4-epsps*, *bar* and *pat*). We speculated that the sample was infected by environmental microorganisms harboring the *HPT* gene often found on processed vegetables.

### 2.2. Isolation and Characterization of the HPT Gene Sequence

To obtain the complete *HPT* gene and flanking sequences, a genome walking PCR strategy was adopted to determine novel and unknown sequences adjacent to *HPT*. A 7259 bp fragment was obtained by overlapping two derived fragments with the *HPT* gene. BLAST analysis revealed that 7259 bp of the isolated sequence was highly homologous to plasmid DNA found in *Escherichia coli*, *Klebsiella pneumoniae*, and *Salmonella* sp., with sequence identity of about 98.1%. Ten plasmid DNA sequences were then collected from Genebank: six *Klebsiella pneumoniae* strains MH15-269 (Accession No. AP023338.1), AR0125 (Accession No. CP021858.1), AR0113 (Accession No. CP021755.1), AR0112 (Accession No. CP021547.1), AR0115 (Accession No. CP021547.1), and CAV1596 (Accession No. CP011644.1); three *Escherichia coli* strains TD-33 (Accession No. MN915013.1), GD-33 (Accession No. MN915011.1), TJ-33 (Accession No. MN915010.1); and one *Salmonella* sp. (Accession No. CP071694.1). Homologous sequence alignment indicated that these conserved regions exhibited some single nucleotide polymorphisms (SNP) (Appendix A). Although *HPT* genes from the plasmid DNA of microorganisms and the isolated sequence had some SNPs (Appendix A), they simultaneously all still probably encoded hygromycin B phosphotransferase (Appendix A).

### 2.3. Design of Primers and Probes

Based on this sequence alignment, a conserved fragment was found to be present in all collected sequences from microorganisms in this study. Therefore, multiple candidate primer/probe sets were designed based on these conserved segments without any SNPs, targeting the junction site between the 5′-plasmid DNA sequence and the *HPT* gene. This segment was then constructed into a pFP-hpt plasmid. The forward primer bound to 5′-plasmid DNA, while the reverse primer binding site was in the *HPT* gene adjacent to the plasmid DNA. The probe bound to the junction between the 5′-plasmid DNA sequence and the *HPT* gene fragment. The locations of primers and probes for qualitative and qPCR methods are shown in Figure 1, and details of primers and probes can be found in Appendix A. To select the best primer/probe set, all possible primer and probe combinations were tested for application size and specificity using 500 copies of pFP-hpt as a template. The most efficient, reliable, and stable qualitative/quantitative PCR primer/probe sets were Ghpt-F3/Ghpt-R3 and Ghpt-QF3/Ghpt-QR3/Ghpt-QP, yielding 261 bp and 156 bp size products, respectively (Appendix A).

### 2.4. Specificity and Sensitivity Test of Singleplex Qualitative

The sequence of the primer set of Ghpt-F3/Ghpt-R3, generating a 261 bp amplicon product, was next analyzed using BLAST. The sequence was mainly homologous to the plasmid DNA found in *E. coli*, *K**. pneumoniae, Salmonella* sp., and the genomic DNA of *Proteus mirabilis*, with 100% identity (Appendix A). No sequence homology was observed with related non-target organisms, such as plant viruses or crop plants. The amplification specificity of the primer set Ghpt-F3/Ghpt-R3 was tested using genomic DNA from non-transgenic plants (*Arabidopsis thaliana*, soybean, maize, peanuts, rice, rapeseed, cotton, wheat, sugar beet, and tomato), GM rice harboring *HPT* gene (Kefeng 6, Kemingdao and SD rice), GM rice (TT51-1 and G6H1), GM soybean mixed sample, GM maize mixed sample, GM rapeseed mixed sample, GM cotton mixed sample, and plasmid DNA (pFP-hpt). The results are presented in Table 1. Plant endogenous genes from 18S rRNA [17] have been observed in specific amplification bands. However, the 261 bp target fragment and 472 bp of *HPT* gene from GM crops have not been observed from plant species, GM rice (TT51-1 and G6H1), GM soybean mixed sample, GM maize mixed sample, GM rapeseed mixed sample, and GM cotton mixed sample in similar assays. Electrophoresis showed that a unique 261 bp fragment was amplified from pFP-hpt plasmid, but there was no amplification observed from Kefeng 6, Kemingdao, and SD rice (Figure 2a). Specificity tests indicated qualitative PCR amplification using the primer set Ghpt-F3/Ghpt-R3 could reliably and specifically detect micro-organisms carrying the *HPT* gene.

In practice, DNA extracted from GM food or feed tends to be commonly infected or contaminated with trace levels of microorganisms. To evaluate the sensitivity of our new qualitative assay system, pFP-hpt plasmid DNA was serially diluted from 500, 100, 50, 20, and 10 copies per microliter and used as a template for PCR analysis. Specific products at 261 bp could be observed in reactions with as few as 20 copies of pFP-hpt as a template (Figure 2b). These results indicate that the detection sensitivity of the qualitative method established in this research was as low as 20 copies.

### 2.5. Determination of the Applicability of Qualitative PCR

To verify the applicability of this new qualitative PCR method, a routine testing sample was numbered 2519, set as a blind sample, and investigated in detail. pFP-hpt plasmid and SD rice were used as positive and negative controls, respectively. The material consisting of extracted rice seed was determined to be positive for the presence of the *HPT* gene, yielding a 472 bp product (Figure 3a). However, it could not be assigned to a GM plant event due to the absence of other screening elements or GM-specific events. Therefore, the material was analyzed with conventional PCR using the primer set Ghpt-F3/Ghpt-R3. Specific 261 bp products from rice sample 2519 and pFP-hpt were observed by electrophoresis (Figure 3b). To confirm the specificity of fragment, the obtained PCR product was sequenced using both forward and reverse primers from this qualitative assay. The sequence alignment of the 261 bp amplicon showed 100% homology with major microbes from *E*. *coli*, *K*. *pneumoniae*, *Salmonella* sp., and *Proteus mirabilis*, which carried the complete *HPT* gene in a plasmid or genomic DNA by further alignment analysis (Appendix A). These results reveal that the qualitative PCR assay established in this research could be applied to exclude false positives for the *HPT* gene during routine GM analysis.

### 2.6. Verification of qPCR Specificity

The sequence of primer/probe set Ghpt-QF3/Ghpt-QR3/Ghpt-QP was next evaluated for use in qPCR. Based on the above analysis, this 156 bp fragment was specific. The experimental specificity of qPCR assay was tested in the same way as that of our qualitative PCR method. There were no positive signals observed in the plant species, GM rice (TT51-1 and G6H1), GM soybean mixed sample, GM maize mixed sample, GM rapeseed mixed sample, and GM cotton mixed sample using qPCR detection for Ghpt-QF3/Ghpt-QR3/Ghpt-QP and the *HPT* gene (Table 1). Only pFP-hpt plasmid DNA was amplified with the primer/probe set Ghpt-QF3/Ghpt-QR3/Ghpt-QP, and no positive signals were observed in GM rice harboring *HPT* gene (Kefeng 6, Kemingdao, and SD rice). These results indicate that qPCR could be used to specifically detect microorganisms carrying the *HPT* gene.

### 2.7. Standard Curve and Repeatability of qPCR

To evaluate the amplification efficiency and linearity of our qPCR assay, a 10-fold series of dilutions of plasmid-pFP-hpt was prepared, corresponding to 500,000, 50,000, 5000, 500, and 50 copies/µL. The serial dilutions were used to establish a standard curve, and each dilution was assayed in triplicate with three parallel reactions. Standard curves were created by plotting Ct values against the logarithm of pFP-hpt plasmid copy numbers. There was a good agreement observed between the quantity of the template and the Ct values, with R^2^ values between 0.999 and 1.000. The slope of the standard curve was calculated between −3.27 to −3.369, with corresponding amplification efficiencies of 98.10% to 99.70% (Figure 4, Table 2). These data met the minimum performance requirements for analytical methods defined by the European Network of GMO Laboratories [18].

The repeatability of the qPCR system was analyzed using these standard curves as described above. With the reduction in plasmid template from 500,000 to 50 copies, the average Ct values of the real-time assay increased from 21.41 to 34.77, with RSD values ranging from 0.24% to 0.99%, and the relative repeatability standard deviation (RSD_r_) ranging from 0.03% to 0.30% (Table 3). These results also comply with the minimum performance requirements for analytical methods defined by the European Network of GMO Laboratories [18] and reveal that the quantitative real-time PCR assay had good stability and reliability.

### 2.8. Limits of Detection and Quantification with qPCR

To determine the sensitivity of this qPCR method, the LOD and LOQ were next determined. The pFP-hpt plasmid was diluted to 50, 20, 10, 5, and 1 copies for analysis of 9 replicate real-time PCR reactions. The detection capacity of our quantitative assay decreased with decreasing template copy numbers (Table 4). All nine PCR replicates had typical amplification curves when the template copy number was 5, whereas only five reactions were positive when the template copy number was 1. Therefore, the LOD of this qPCR method was estimated to be five copies of plasmid. Relative repeatability standard deviation (RSD_r_) of Ct values from the nine replicates were all below 25% with the reduction in the template copy number. The LOQ of our method was approximately five copies of plasmid based on the implementation of the guidance document of the Joint Research Centre, Institute for Reference Materials and Measurements (JRC-IRMM). These results indicate that singleplex qPCR could be used to detect micro-organism infection carrying the *HPT* gene with high sensitivity but should not be used to accurately quantify the amount of infection.

### 2.9. Analysis of HPT-Positive Seed Material Using qPCR

The rice sample 2519 was also used to verify the applicability of our qPCR method. The pFP-hpt plasmid, non-transgenic rice, and sterile water were used as a positive control, negative control, and no template control, respectively. The results show that rice 2519 had a typical amplification curve with Ct values of less than 30, which correspond to approximately 500 copies of plasmid from an infected bacterium (Figure 5). The results of the 156 bp PCR product alignment are consistent with those of the 261 bp sequence produced using our qualitative method, as indicated by BLAST analysis. These results reveal that the qPCR assay developed in this paper could also be applied to exclude false positives for the *HPT* gene during routine GM analysis.

### 2.10. Development of a Qualitative and Quantitative Duplex PCR

The specificity of the designed primers/probes above were assessed using a singleplex PCR assay, as the conventional and real-time PCR methods that have been used in the field to detect the *HPT* gene were singleplex and tested for specificity [19]. Considering the time and cost of analysis, we developed a qualitative and quantitative duplex PCR test for simultaneous detection of the *HPT* gene from possible GM introduction and environmental micro-organisms harboring the *HPT* gene. Figure 6 shows the results of a qualitative duplex PCR with rice sample 2519 infected with environmental microorganisms. We have also developed a quantitative duplex test utilizing the two specific sets of primers and probes, one for the *HPT* gene and Ghpt-QF3/Ghpt-QR3/Ghpt-QP for the detection of environmental micro-organisms harboring the *HPT* gene (Table 5). Consequently, the combination of both targets, the *HPT* gene and environmental micro-organisms harboring the *HPT* gene, was viable in one PCR reaction system.

## 3. Discussion

Due to its extensive use as selectable marker gene in the first generation of GMOs, the *HPT* gene derived from *E. coli* has become the generally used PCR target for screening tests in routine GMO analysis [3]. In this study, only the *HPT* gene was detected from soybean sprouts in a routine test in our laboratory. For further analysis of whether this sample was a non-authorized GMO, the genome walking strategy was used to identify unknown sequences based on the *HPT* gene. Eventually, a 7259 bp sequence containing the *HPT* gene was retrieved after multiple rounds of PCR amplification. BLAST analysis revealed that this 7259 bp plasmid fragment might have derived from strains of *E. coli*, *K**. pneumoniae* or *Salmonella* sp. The findings suggest that the presence of environmental micro-organisms in soybean sprouts resulted in the presence of the *HPT* gene in the samples. Determining whether the *HPT* gene is derived from genetically modified crops or nearby micro-organisms will require reliable and stable detection methods.

In recent years, transgenic testing laboratories have paid increasing attention to infection and cross-contamination during GMO detection from food, feed, and seeds. Formerly, conventional PCR methods based on ORF VI and ORF V were used to detect CaMV to distinguish P35S derived from CaMV or transgenic crops [4]. Real-time PCR and multiplex PCR methods targeting different conserved regions were designed to enhance the accuracy and coverage of CaMV detection [5,6,7,8]. Real-time PCR methods for the detection of figwort mosaic virus (FMV) were developed to complement the FMV 34S promoter-specific PCR assay used for the screening of GMOs in food and feed [9]. In addition, a QRT-PCR assay targeting the TMS1 region was successfully applied to distinguish elements such as T-nos, P-nos, and T-3′g7, which have been introduced by *Agrobacterium* sp. into GMOs plant events from natural *Agrobacterium* sp. [20].

Here, through integrating a 3207 bp sequence into a plasmid vector, we developed a novel plasmid as a positive sample, pFP-hpt, carrying 285 bp of the incomplete *HPT* gene and 2922 bp of the 5′ flanking sequence. A specific qualitative and qPCR assay targeting 261 bp and 156 bp between the *HPT* gene and its 5′flanking adjacent sequence, which were conserved sections without SNPs by limited alignment analysis, were established to complement *HPT* gene assays used for screening first generation GMOs. These qualitative and qPCR tests showed high specificity and sensitivity, even when a trace amount of the *HPT* gene from micro-organisms was present. Combining this with our previously validated *HPT* gene method [16], these assays can be applied to routine testing in the laboratory to avoid false positives of the *HPT* selectable marker. The LOD and LOQ of our qPCR assay were estimated to be five copies of plasmid DNA. In this paper, we demonstrated that the pFP-hpt plasmid prepared in this study can be used as an alternative standard for the quantification of the *HPT* gene from natural occurrences.

Generally, GMO detection is divided into two steps: element-screening PCR is often the initial step in GMO testing procedures. When results indicate the presence of specific genetic elements, identification or quantification of the GM plant is followed by event-specific assays [21]. This two-step detection procedure requires a large amount of time and reagents. The developed qualitative and quantitative duplex PCR could detect two targets in one PCR reaction at the same time, thereby reducing the reagent cost, shortening the detection time, improving the detection efficiency, and providing an effective method for the rapid detection of the *HPT* gene and micro-organisms harboring the *HPT* gene. This is the first study to attempt to avoid *HPT* gene false-positive tests derived from bacterial plasmids, and future work should be conducted on bacterial genomes carrying the *HPT* gene to improve false-positive exclusive detection systems. We will also expand the application of this method for daily detection in transgenic testing laboratories.

## 4. Materials and Methods

### 4.1. Materials

Soybean sprouts used for GMO screening and genome walking PCR were purchased from a local market in Wuhan. *A**. thaliana*, sugar beet (*Beta vulgaris*), cotton (*Gossypium hirsutum*), soybean (*Glycine max*), rice (*Oryza sativa*), wheat (*Triticum aestivum*), maize (*Zea mays*), peanuts (*Arachis hypogaea*), rapeseed (*Brassica napus*), and tomato (*Lycopersicon esculentum* Mill) were used for specificity testing and were collected by our lab. The transgenic rice lines used in this study were TT51-1, G6H1, Kefeng 6, Kemingdao, and SD rice [22]. A GM soybean mixed sample (GTS40-3-2, MON89788, A5547-127, A2704-12, 356043, 305423, CV127, MON87701, MON87708, MON87769, MON87705, FG72, DAS81419-2; 1% (*w*/*w*) for each variety), GM maize mixed sample (Bt11, Bt176, MON810, MON863, GA21, NK603, T25, TC1507, MON89034, MON88017, 59122, MIR604, 3272, MON87460, DAS40278-9, 4114, MON87427, 5307; 1% (*w*/*w*) for each variety), GM cotton mixed sample (MON1445, MON531, MON15985, LLCOTTON25, MON88913, GHB614, COT102; 1% (*w*/*w*) for each variety), and GM rapeseed mixed sample (MS1, MS8, RF1, RF2, RF3, T45, Oxy-235, Topas19/2, MON88302, 73496; 1% (*w*/*w*) for each variety) were provided by Development Center of Science and Technology, MARA. The routine testing sample for rice seed (2519) used in this research is commonly available.

### 4.2. DNA Extraction and Purification

For GMO screening of all samples and genome walking, genomic DNA was isolated and purified using the DNeasy Plant Maxi Kit (QIAGEN, Hilden, Germany) according to the manufacturer’s protocol. For qualitative and quantitative method in-house validation, genomic DNA was extracted and purified from young leaves or seeds of species plants using a cetyltrimethylammonium bromide (CTAB)-based protocol [23]. Plasmid DNA was prepared using the TIANprep mini plasmid kit (TIANGEN, Beijing, China) following the manufacturer’s recommendations. DNA concentrations were analyzed using a NanoDrop ONE^c^ (Thermo Scientific, Waltham, MA, USA) spectrophotometer.

### 4.3. Genome Walking PCR

The complete *HPT* gene and its flanking sequence were isolated utilizing the GenomeWalker Universal Kit (Clontech, Mountain View, CA, USA) according to the manufacturer’s instructions. Gene-specific primers (GSPs) were designed according to the sequence of the *HPT* gene (Accession No. AF234296) and these isolated sequences. The primers designed for this study are listed in Appendix A.

For genome walking, the first round of Genome Walker PCR was performed with primer sets wHPT1-1/AP1 and wHPT1-2/AP2, generating a fragment containing an incomplete *HPT* gene and a 5′-upstream unknown sequence adjacent to the *HPT* gene. Four successive rounds of Genome Walker PCR were carried out to isolate upstream unknown sequences, using primer pairs wHRSUW1-1/AP1 and wHRSUW1-2/AP2, wHRSUW2-1/AP1 and wHRSUW2-2/AP2, wHRSUW3-1/AP1 and wHRSUW3-2/AP2, and wHRSUW4-1/AP1 and wHRSUW4-2/AP2. The single, major PCR products were directly sequenced with the nested primers after purification by Sangon Biotech (Shanghai, China). The isolated sequences were verified by PCR amplification and then spliced using Vector NTI software (VectorNTIadvance11.5.1, Invitrogen, Waltham, MA, USA). A 5295 bp 5′-upstream fragment containing 173 bp of incomplete *HPT* was obtained by sequence assembly. After one round of PCR amplification, a 1306 bp 3′-downstream fragment, including 195 bp of incomplete *HPT*, was isolated using primer sets wHPT1+1/AP1 and wHPT1+2/AP2. Subsequently, a 5295 bp-upstream segment, the *HPT* gene (Accession No. AF234296), and a 1306 bp 3′-downstream segment were spliced using Vector NTI software (VectorNTIadvance11.5.1, Invitrogen, Waltham, MA, USA) to attain a final sequence of 7259 bp (Appendix A). The sequence was then analyzed using the Basic Local Alignment Search Tool (BLAST, https://blast.ncbi.nlm.nih.gov/Blast.cgi; accessed on 7 February 2020) [24]. Nucleotide BLAST was carried out using the blastn program on the GeneBank database.

### 4.4. Construction of a Reference Plasmid

After characterizing the isolated sequence, a 3207 bp sequence (Appendix A) consisting of 285 bp of the *HPT* gene sequence and its 2922 bp 5′ flanking DNA sequence were synthesized by Sangon Biotech and sub-cloned into plasmid vector pUC57, giving rise to the reference plasmid pFP-hpt. pFP-hpt was used for detecting false-positive GM plants due to micro-organism infection or sample contamination.

### 4.5. Primer and Probe Design

Qualitative PCR oligonucleotide primers were designed using the software Primer Premier Version 5.00 (PREMIER Biosoft International, Palo Alto, CA, USA). qPCR primers and TaqMan fluorescent probes were designed using the software Beacon Designer 8.0 (PREMIER Biosoft, USA). All the primers and probes were synthesized by Sangon Biotech (Appendix A).

### 4.6. Qualitative PCR Conditions

Qualitative PCR amplifications were performed using a Bio-rad C1000 thermal cycler using an optimized PCR mixture including: 10× Taq Buffer, 2.5 mM MgCl_2_, 200 µM of each dNTP, 0.4 µM of each primer, 0.7 unit Taq DNA Polymerase (TIANGEN, Beijing, China), and 50 ng of genomic DNA or plasmid DNA. Then, 500 copies, 100 copies, 50 copies, 20 copies, or 10 copies were used as templates in a final volume of 25 µL.

In the duplex qualitative PCR assay, the reaction mixture (25 µL) contained 10× Taq Buffer, 2.5 mM MgCl_2_, 200 µM of each dNTP, 0.4 µM of, 0.4 µM of Ghpt-F3/Ghpt-R3, 0.7 units Taq DNA Polymerase (TIANGEN, Beijing, China), and 50 ng of genomic DNA or plasmid DNA (500 copies) as a template.

The PCR used the following program: a 95 °C initial denaturation step for 5 min; followed by 35 cycles of 95 °C denaturation for 30 s, 58 °C annealing for 30 s, and 72 °C extension for 30 s; and a 72 °C final extension for 7 min. The PCR products were electrophoresed on 2% agarose gels in 1× TAE buffer and stained with Ultro GelRed (Vazyme, Nanjing, China) for visualization.

### 4.7. qPCR Conditions

qPCR amplification was performed on a CFX96 Real-Time System (Bio-Rad Laboratories, Hercules, CA, USA). Fluorescence signals were monitored using CFX Manager ver. 1.6 (Bio-Rad). Data were analyzed using the CFX Manager ver. 1.6 (1.6.541.1028) software (Bio-Rad).

qPCR assay was performed in an optimized reaction system of 20 µL containing: 10 µL of 2× Premix Ex Taq (probe qPCR) (Takara, Kusatsu, Shiga, Japan), 0.4 µM primers, 0.2 µM probe, and 100 ng of genomic DNA or 5.2 × 10^3^ copies of plasmid DNA as a template. The TaqMan probes used in these qPCR experiments were labelled with the fluorescent reporter 6-carboxy-fluorescein (FAM) and the non-fluorescent Black Hole Quencher 1 (BHQ1).

In the duplex qPCR assay, the reaction mixture (20 µL) contained 2× Premix Ex Taq (probe qPCR) (Takara, Kusatsu, Shiga, Japan), 0.4 µM of qHptF286/qHptR395 and Ghpt-QF3/Ghpt-QR3, 0.2 µM of HptR697 and Ghpt-QP, and 50 ng of genomic DNA or plasmid DNA (500 copies) as a template. The probe for HptR697 was labelled with Hexachlorofluorescein (HEX) and BHQ1, and the probe for Ghpt-QP was labelled with FAM and BHQ1.

All reactions were tested in triplicate with the same program: an initial denaturation step for 3 min at 95 °C, followed by 40 cycles or 50 cycles of 5 s at 95 °C (denaturation) and 1 min at 60 °C (annealing and extension). The fluorescence signal was measured after each annealing and extension step.

### 4.8. Specificity Tests

To evaluate the specificity of our qualitative and qPCR assays, non-GM plants, such as *A**. thaliana*, soybean, maize, peanuts, rice, rapeseed, cotton, wheat, sugar beet, tomato, GM soybean mixed sample, GM maize mixed sample, GM rapeseed mixed sample, GM cotton mixed sample, GM rice including TT51-1, G6H1, Kefeng 6, Kemingdao, and SD rice, as well as a reconstructed plasmid for pFP-hpt, were all tested. The assay was considered specific when the anticipated amplification was detected for each sample.

### 4.9. Sensitivity Tests

To determine sensitivity of our qualitative PCR assay, the serial dilutions of the pFP-hpt plasmid from 500 copies to 10 copies were tested for each target. Two parallel reactions were performed.

To evaluate the LOD and LOQ for our qPCR assay, a plasmid DNA series with copy numbers from 50, 20, 10, 5, and 1 was used. Three parallel reactions were performed on nine replicates of PCR runs. The LOD was analyzed using the cycle threshold (Ct) values of each reaction and all replicates could be reliably detected. The LOQ was estimated as the lowest copy number with the relative standard deviation (RSD) of all replicates below 25% for the measured copy number in this assay.

## 5. Conclusions

In summary, the quantitative PCR and qPCR system demonstrated here was more sensitive and specific for the *HPT* of a bacterial donor during routine GM testing of materials such as seeds. Even though the frequency of *HPT* false-positive test results is not currently known, the methods here allow GMO testing laboratories to distinguish between the *HPT* gene from GM plants and naturally occurring bacteria.

## Figures and Tables

**Figure 1 ijms-23-10000-f001:**
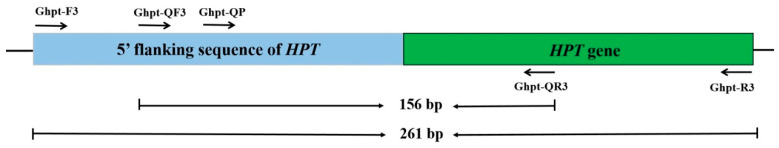
Location of primer/probes on the sequence for qualitative and quantitative PCR methods for false-positive detection.

**Figure 2 ijms-23-10000-f002:**
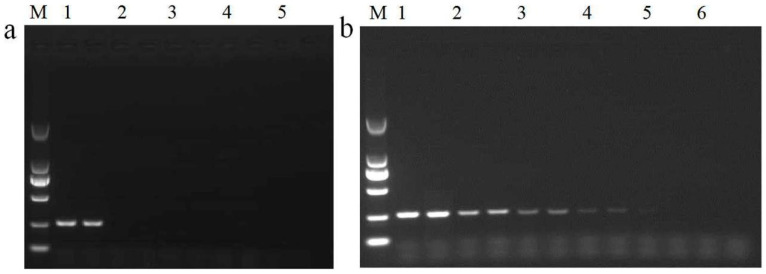
Assessment of the specificity and sensitivity of *HPT* qualitative PCR detection. (**a**) Testing of the amplification specificity of qualitative PCR. Lane M: DL 2000 DNA Marker. Lane 1: pFP-hpt plasmid, Lane 2: Kefeng6, Lane 3: Kemingdao, Lane 4: SD rice, and Lane 5: no template control; each template was run with two parallel reactions. (**b**) Sensitivity of the qualitative PCR assay. Lane M: DL 2000 DNA Marker. Lanes 1–6 correspond to 500, 100, 50, 20, 10, and 0 copies of pFP-hpt, respectively; each template was run with two parallel reactions.

**Figure 3 ijms-23-10000-f003:**
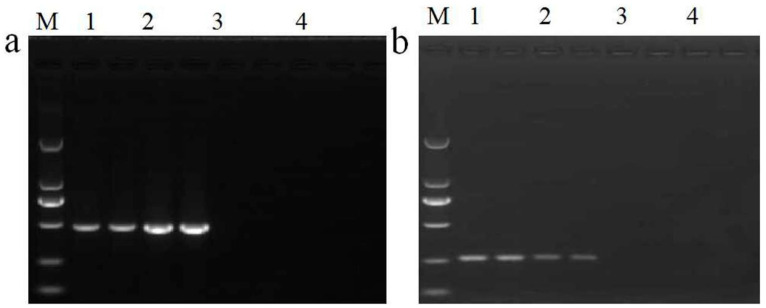
Assessment of applicability for qualitative PCR detection. (**a**) Screening of *HPT* gene in routine samples. Lane M: DL 2000 DNA Marker. Lane 1: routine testing rice material 2519, Lane 2: SD rice as a positive control, Lane 3: pFP-hpt plasmid, and Lane 4: no template control; each template was run with two parallel reactions. (**b**) False-positive detection using primer set Ghpt-F3/Ghpt-R3. Lane M: DL 2000 DNA Marker. Lane 1: blind rice 2519, Lane 2: pFP-hpt plasmid as a positive control, Lane 3: SD rice, and Lane 4: no template control; each template was run with two parallel reactions.

**Figure 4 ijms-23-10000-f004:**
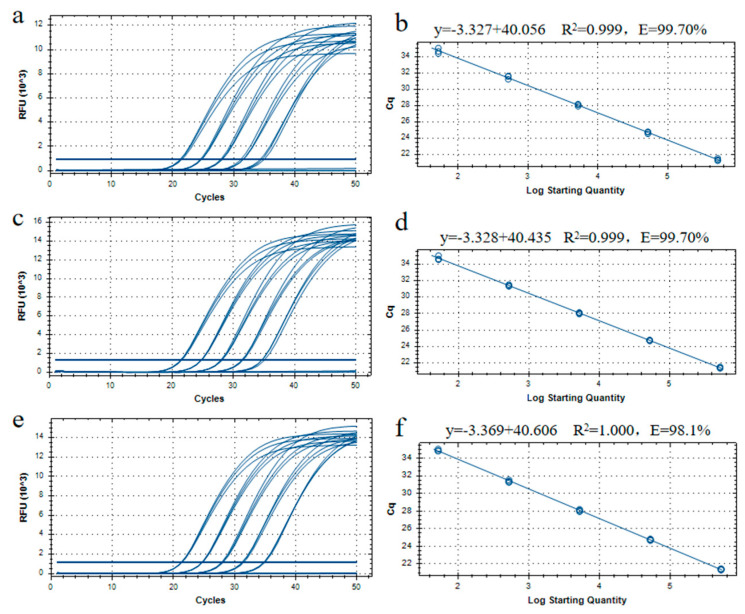
Amplification and standard curves for the quantitative PCR method using gradient-diluted pFP-hpt plasmid DNA as a template. (**a**,**c**,**e**): Amplification graph for the quantitative PCR assay using primer/probe set Ghpt-QF3/Ghpt-QR3/Ghpt-QP with three repetitions. (**b**,**d**,**f**) Standard curves for the quantitative PCR assay corresponding to (**a**,**c**,**e**), respectively. The copy numbers of pFP-hpt were 500,000, 50,000, 5000, 500, and 50 copies in each reaction.

**Figure 5 ijms-23-10000-f005:**
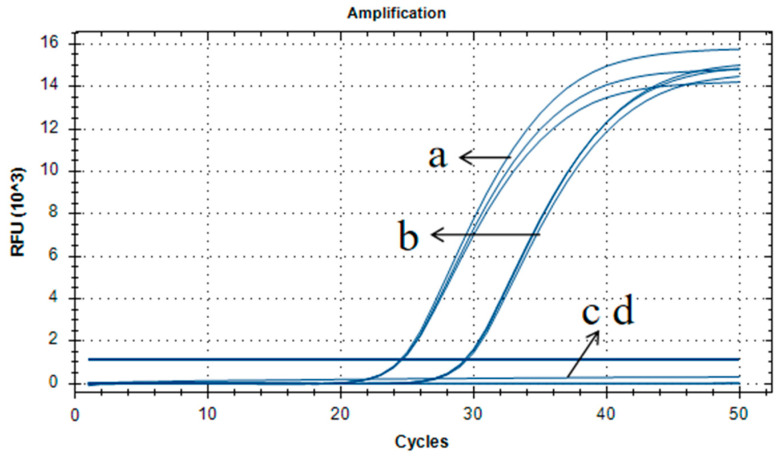
Assessment of applicability for qPCR detection using primer/probe set of Ghpt-QF3/Ghpt-QR3/Ghpt-QP. (a–d) Amplification curve for pFP-hpt, routine testing rice material 2519, non-transgenic rice, and no template control, respectively.

**Figure 6 ijms-23-10000-f006:**
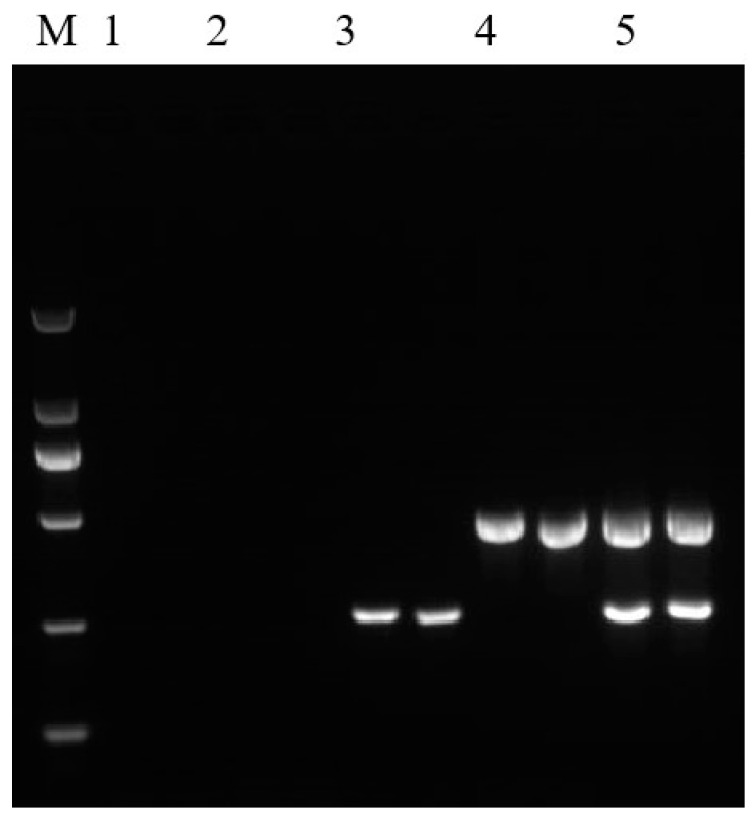
Results of qualitative duplex PCR amplification. Lane M: DL 2000 DNA Marker. Lane 1: no template control, Lane 2: non-GM rice, Lane 3: pFP-hpt plasmid, Lane 4: Kemingdao, and Lane 5: routine testing rice material 2519; each template was run with two parallel reactions.

**Table 1 ijms-23-10000-t001:** Results of specificity tests for qualitative PCR and qPCR assays from different species of plants.

Target Species Plant	18S rRNA	Qualitative PCR		qPCR	
		Ghpt-F3/R3	*HPT* Gene	Ghpt-QF3/QR3/QP	*HPT* Gene
*Arabidopsis thaliana*	+	−	−	−	−
Soybean (*Glycine max*)	+	−	−	−	−
Maize (*Zea mays*)	+	−	−	−	−
Peanuts (*Arachis hypogaea*)	+	−	−	−	−
Rice (*Oryza sativa*)	+	−	−	−	−
Rapeseed (*Brassica napus*)	+	−	−	−	−
Cotton (*Gossypium hirsutum*)	+	−	−	−	−
Wheat (*Triticum aestivum*)	+	−	−	−	−
Sugar beet (*Beta vulgaris*)	+	−	−	−	−
Tomato (*Lycopersicon esculentum* Mill)	+	−	−	−	−
GM soybean mixed sample	+	−	−	−	−
GM maize mixed sample	+	−	−	−	−
GM rapeseed mixed sample	+	−	−	−	−
GM cotton mixed sample	+	−	−	−	−
TT51-1	+	−	−	−	−
G6H1	+	−	−	−	−
Kefeng6	+	−	+	−	+
Kemingdao	+	−	+	−	+
SD rice	+	−	+	−	+

“+,” positive amplification; “−,” no amplification.

**Table 2 ijms-23-10000-t002:** Repeatability of characteristic parameters of standard curves from qPCR.

Repeat	Slope	Mean	SD	RSD (%)	R^2^	Mean	SD	RSD (%)	Efficiency	Mean	SD	RSD (%)
1	−3.327	−3.341	0.024	0.717	0.999	0.999	0.001	0.058	99.70%	99.17%	0.009	0.932
2	−3.328				0.999				99.70%			
3	−3.369				1.000				98.10%			

SD, standard deviation; RSD, relative standard deviation; R^2^, coefficient of determination.

**Table 3 ijms-23-10000-t003:** Repeatability of Ct values of standard curves from qPCR.

Copy Number	Repeat	Ct Value			Mean of Ct Values	SD	RSD (%)	Mean of All Ct Values	SD_r_	RSD_r_ (%)
		1	2	3						
500,000	1	21.28	21.59	21.40	21.42	0.16	0.73	21.41	0.03	0.15
	2	21.34	21.52	21.42	21.43	0.09	0.42			
	3	21.30	21.44	21.37	21.37	0.07	0.33			
50,000	1	24.58	24.82	24.77	24.72	0.13	0.51	24.73	0.01	0.03
	2	24.68	24.80	24.73	24.74	0.06	0.24			
	3	24.66	24.81	24.73	24.73	0.08	0.30			
5000	1	27.95	28.20	28.14	28.10	0.13	0.46	28.06	0.03	0.11
	2	27.94	28.13	28.04	28.04	0.10	0.34			
	3	27.94	28.17	28.06	28.06	0.12	0.41			
500	1	31.24	31.64	31.58	31.49	0.22	0.69	31.42	0.06	0.18
	2	31.37	31.52	31.31	31.40	0.11	0.34			
	3	31.35	31.52	31.26	31.38	0.13	0.42			
50	1	34.64	34.38	35.06	34.69	0.34	0.99	34.77	0.11	0.30
	2	34.62	35.04	34.54	34.73	0.27	0.77			
	3	35.03	34.82	34.83	34.89	0.12	0.34			

Ct, cycle threshold.

**Table 4 ijms-23-10000-t004:** Estimation of LOD and LOQ of qPCR using serially diluted plasmid DNA.

Plasmid Copy Number	Repeat	Ct Value			Mean of Ct Values	SD	RSD (%)	Mean of All Ct Values	SD_r_	RSD_r_ (%)
		1	2	3						
50	1	34.79	34.45	34.32	34.52	0.24	0.70	34.59	0.06	0.17
	2	34.87	34.68	34.32	34.62	0.28	0.81			
	3	34.46	35.14	34.25	34.62	0.47	1.34			
20	1	35.75	35.62	35.32	35.56	0.22	0.62	36.18	0.55	1.51
	2	36.81	36.11	36.92	36.61	0.44	1.20			
	3	36.05	36.25	36.76	36.35	0.37	1.01			
10	1	37.09	37.07	36.65	36.94	0.25	0.67	36.75	0.34	0.94
	2	35.76	36.97	36.34	36.36	0.61	1.66			
	3	38.77	36.34	35.79	36.97	1.59	4.29			
5	1	37.32	37.32	40.58	38.41	1.88	4.90	38.36	0.23	0.60
	2	38.1	38.57	37.68	38.12	0.45	1.17			
	3	37.83	40.41	37.47	38.57	1.60	4.16			
1	1	38.34	38.46	40.31	39.04	1.10	2.83	39.06	/	/
	2	NA	38.71	NA	38.71	/	/			
	3	NA	39.44	NA	39.44	/	/			

NA, no amplification; /, no data.

**Table 5 ijms-23-10000-t005:** Results from duplex qPCR amplification.

Samples	Ct Values of the *HPT* Gene			Ct Values of Microorganisms Assay		
	1	2	3	1	2	3
no template control	NA	NA	NA	NA	NA	NA
non-GM rice	NA	NA	NA	NA	NA	NA
plasmid of pFP-hpt	NA	NA	NA	26.83	26.77	27.03
Kemingdao	23.88	23.74	23.86	NA	NA	NA
Routine testing rice 2519	28.62	28.54	28.61	26.91	26.91	27.03

NA, no amplification.

## Data Availability

Not applicable.

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
