# Peer review of "Qualitative and Quantitative Real-Time PCR Methods for Assessing False-Positive Rates in Genetically Modified Organisms Based on the Microbial-Infection-Linked HPT Gene"

_ijms, 2022, doi:10.3390/ijms231710000_

Round 1

Reviewer 1 Report

Summary

The authors developed new testing assays that can detect specific microbial infections that can trigger false positive results in common GMO testing assays. The assays could be used in parallel with existing GMO assays in routine testing to diferentiate between bacterial infection and GMO material.

General comments

Is there a risk that GMOs could also additionally contain microorganisms and would be falsely detected as a "false positive"? How does the test procedure account for this? If this is an open challenge it should be mentioned.

If microbiotic infection is common, was there any method used to ensure the plant and GMO samples used were not contaminated with microorganisms? Or would this only have been done posterior when a sample would have tested positive in the assay?

As I understand only one sample (rice 2519) was used to validate the assay with samples that tested (false-)positive for GMO. Do you consider that a limitation? why/why not?

Specific comments

Abstract:

Line 27: QRT-PCR is an unusual abreviation for quantitative real-time PCR. The MIQUE guidlines recommend using "qPCR" which also avoids confusion with reverse-transcription PCR. Consider using MIQUE reporting standard also for the rest of the paper.

Results:

Table 1: Please improve table layout

Table 2/Line 210: Why are the results from the GMO samples not included in the table?

Line 227: (b)(d)(e) should read (b)(d)(f) instead

Table 3: Please improve table layout to avoid line breaks in numbers

Line 235: RSD Value of 0.21 does not exist in Table four. Do you mean 0.24?

Table 5: It is unclear what the meaning of the rows, columns is for the repeats. Is this just for layout purposes to show the values of the 9 replicates? If yes, why are mean, sd, rsd calculates both per row and again for the totals? Also Explain the meaning of "/" in the legend.

Line 262: no temple control => no template control

Figure 4+5 look blurry. Please investigate if it is possible to export the graphs using a vector format or higher resolution.

Table 6: The heading "Ct values of QRT-PCR" is confusing as all values have been generated by QRT-PCR. I think it should read something like "Ct values of microorganisms assay".

Discussion:

Line 384+385: Fix BLAST link/reference

Author Response

Reviewer #1:

Comment 1:Is there a risk that GMOs could also additionally contain microorganisms and would be falsely detected as a "false positive"? How does the test procedure account for this? If this is an open challenge it should be mentioned.

Response: It is a real risk that GMOs could also additionally contain microorganisms. The duplex droplet digital PCR for HPT gene and microorganisms detection can solve this problem. If the GMOs carry microorganisms simultaneously, the copy number of HPT gene is more than the copy number for microorganisms. In the next step, we will try to implement this research to further improve the test procedure.

Comment 2:If microbiotic infection is common, was there any method used to ensure the plant and GMO samples used were not contaminated with microorganisms? Or would this only have been done posterior when a sample would have tested positive in the assay?

Response: Microbiotic infection is common in the natural environment. High temperature processing and cryopreservation are effective ways to prevent GMO samples from being contaminated with microorganisms. Washing samples with sterile water prior to transgenic testing is also a good way to avoid false positive results from microbial infection. When the sample has still been done posterior, it is still necessary to perform a false positive eliminating test.

Comment 3: As I understand only one sample (rice 2519) was used to validate the assay with samples that tested (false-)positive for GMO. Do you consider that a limitation? why/why not?

Response: In our daily testing, we found several rice samples that only tested positive for HPT gene. Then we performed false positive exclusion tests for microbial infection using the qualitative PCR and qPCR assays established in this study. The methods showed the best amplification efficiency to distinguish genetically modified (GM) crops from microbial infection. However, the amplification results of the 2519 sample were only used to demonstrate the applicability of the assays. In the future, we will broaden the application of these assays to routine GMO testing.

Comment 4: Line 27: QRT-PCR is an unusual abreviation for quantitative real-time PCR. The MIQUE guidlines recommend using "qPCR" which also avoids confusion with reverse-transcription PCR. Consider using MIQUE reporting standard also for the rest of the paper.

Response: We have changed the “QRT-PCR” to “qPCR” in the new version thoroughly.

Comment 5: Table 1: Please improve table layout.

Table 3: Please improve table layout to avoid line breaks in numbers

Response: We have improved table layout in the new version thoroughly. Base on reviewer#3’s comment, Table 1 was added to supplementary materials, and reedited as Table S3.

Comment 6: Table 2/Line 210: Why are the results from the GMO samples not included in the table?

Response: In view of the reviewer's comments, we supplemented the experiment as soon as possible. GM rice (TT51-1 and G6H1), GM soybean mixed sample, GM maize mixed sample, GM rapeseed mixed sample, GM cotton mixed sample were also used to evaluate the specificity for the qualitative PCR and qPCR methods. The results are presented in Table 2 and elaborated in the manuscript. Table 2 was reedited as Table 1 in new version.

Comment 7: Line 227: (b)(d)(e) should read instead (b)(d)(f)

Response: Line235: We have corrected “(b)(d)(f) ” instead of (b)(d)(e) in the new version.

Comment 8: Line 235: RSD Value of 0.21 does not exist in Table four. Do you mean 0.24?

Response: This is a typing mistake for the RSD value of 0.21. We have corrected it as 0.24.  

Comment 9: Table 5: It is unclear what the meaning of the rows, columns is for the repeats. Is this just for layout purposes to show the values of the 9 replicates? If yes, why are mean, sd, rsd calculates both per row and again for the totals? Also Explain the meaning of "/" in the legend.

Response: In the LOD and LOQ of qPCR evaluation experiments, we set three replicate experiments with three parallel responses for each replicate. In Table 5, each row means the data of three parallel reactions obtained from a replicate experiment.  "Mean of Ct values" is the average of three parallel data per row. Then intra-replicate SD and RSD were calculated. "Mean of all Ct values" was actually calculated according to three replicates of "Mean of Ct values" per column, which was also the average of 9 replicates. The inter-replicate SDr and RSDr were calculated by column. In the literature, they are commonly expressed as “Mean of Ct values" and “Mean of all Ct values”. We have also recalculated and supplemented the data in Table 5, and explain the meaning of "/" in the legend. Table 5 was reedited as Table 4 in new version.

Comment 10: Line 262: no temple control => no template control

Response: Line 270: We have corrected “no template control”.

Comment 11: Figure 4+5 look blurry. Please investigate if it is possible to export the graphs using a vector format or higher resolution.

Response: We enhanced the resolution of Figure 4 and Figure 5. They look clearer in the new version.

Comment 12: Table 6: The heading "Ct values of QRT-PCR" is confusing as all values have been generated by QRT-PCR. I think it should read something like "Ct values of microorganisms assay".

Response: We have modified "Ct values of QRT-PCR" to "Ct values of microorganisms assay" in table 6. Table 6 was reedited as Table 5 in new version.

Comment 13: Discussion: Line 384+385: Fix BLAST link/reference

Response: I think this review comment should refer to line 404 in 4.3 Genome walking PCR. I have fixed the BLAST link and added the reference in the new version.

Reviewer 2 Report

Comments#

 In the article entitled “Qualitative and quantitative real-time PCR methods for assessing false-positive rates in genetically modified organisms based on the microbial infection linked HPT gene”, Li et.al developed duplex PCR methodology (qualitative PCR and QRT-PCR) to identify the presence of the marker gene- HPT (Hygromycin phosphotransferase)

in the genetically modified plants and therefore were able to differentiate between GM crops and crops infected with microbes harboring the HPT genes. The current approach eliminates the possible false-positive results during the detection of GM plant samples.

The experiment is scientifically sound, and the manuscript is easy to read and understand. However, please consider revising the manuscript thoroughly for the proper usage of scientific names.

Minor comments#

Line 24 - Change ‘genome walker’ to ‘genome walking’

Line 61 – Change ‘natural occurring’ to ‘naturally occurring'

Lines 110-117 – Italicize the scientific names.

Line 119-122 – It is true that SNPs do not impact the gene functions in all scenarios. However, please provide evidence for this statement – sequence alignment indicating the SNPs as supplementary material.

Lines 146-147 - Italicize the scientific names.

Line 299 – Change to ‘genome walking strategy’

Author Response

Reviewer #2:

Comment1:Line 24 - Change ‘genome walker’ to ‘genome walking’

Line 299 – Change to ‘genome walking strategy’

Response: We have made the change in the new version thoroughly.

Comment2:Line 61 – Change ‘natural occurring’ to ‘naturally occurring'

Response: We have made the change in the new version.

Comment3:Lines 110-117 – Italicize the scientific names.

Lines 146-147 - Italicize the scientific names.

Response: We have italicized the microorganisms names, and revised the manuscript thoroughly in new version.

Comment4:Line 119-122 – It is true that SNPs do not impact the gene functions in all scenarios. However, please provide evidence for this statement – sequence alignment indicating the SNPs as supplementary material.

Response: We have provided the sequence alignment information in supplementary material as Figure S2 and Figure S3.

Reviewer 3 Report

I appreciate the opportunity to review this manuscript.

The subject discussed is of great importance in laboratory evaluations.

Congratulations for the initiative.

Author Response

Reviewer #3:

Since the reviewer did not list comments, and gave them directly in the manuscript using the endorsement mode. our responses are as follows.

Response1: In new version, we have removed “E. coli” in line 26.

Response2: Italicssp from Salmonella sp. and “Agrobacterium sp” were revised to “sp” in the whole manuscript.

Response3: We have revised abreviation for quantitative real-time PCR to “qPCR” according the MIQUE guidlines. The reviewer#1 also mentioned this comment.  

Response4: We have italicized the microorganisms names, HPT gene, Arabidopsis thaliana. And we revised the manuscript thoroughly in new version.

Response5: In the original manuscript, Table 1 was added to supplementary materials, and reedited as Table S3.

Response6: We have centered the Figure 2, Figure 3, Figure 4 and Figure 5.

Response7: In new version, we have modified format of “Lane3” to “Lane 3” in Line 201.

Response8: We have replaced “Klebsiella pneumoniae by “K. pneumoniae”in new version. And “Arabidopsis thaliana” was replace by “A. thaliana” in Materials and Method.